# Can 3D RVEF be Prognostic for the Non-Ischemic Cardiomyopathy Patient but Not the Ischemic Cardiomyopathy Patient? A Cardiovascular MRI Study

**DOI:** 10.3390/diagnostics9010016

**Published:** 2019-01-23

**Authors:** Sartaj S. Gill, Mark Doyle, Diane V. Thompson, Ronald Williams, June Yamrozik, Saundra B. Grant, Robert W. W. Biederman

**Affiliations:** 1Department of Medicine, Cardiology Division, Morehouse School of Medicine, Atlanta, GA 30310, USA; sarvgill@yahoo.com; 2Gerald McGinnis Cardiovascular Institute, Allegheny General Hospital, Pittsburgh, PA 15212, USA; mdoyle@wpahs.org (M.D.); vido@wpahs.org (D.V.T.); rwillia@wpahs.org (R.W.); jyamrozi@wpahs.org (J.Y.); sgrant@wpahs.org (S.B.G.)

**Keywords:** right ventricular ejection fraction, prognosis, heart failure, non-ischemic cardiomyopathy, cardiac MRI

## Abstract

Background: While left ventricular ejection fraction (LVEF) has been shown to have prognostic value in ischemic cardiomyopathy (ICMX) patients, right ventricular ejection fraction (RVEF) has not been systematically evaluated in either ICMX or non-ischemic cardiomyopathy (NICMX) patients. Moreover, an accurate estimation of RVEF is problematic due to the geometry of the right ventricle (RV). Over the years, there have been improvements in the resolution, image acquisition and post-processing software for cardiac magnetic resonance imaging (CMR), such that CMR has become the “gold standard” for measuring RV volumetrics and RVEF. We hypothesize that CMR defines RVEF more so than LVEF and might have prognostic capabilities in ischemic and non-ischemic cardiomyopathy patients (ICMX and NICMX). Methods: Patients that underwent CMR at our institution between January 2005 and October 2012 were retrospectively selected if three-dimensional (3D) LVEF < 35%. Patients were further divided into ICMX and NICMX groups. The electronic medical record (EMR) database inquiry determined all-cause mortality and major adverse cardiovascular events (MACE). Additionally, a Social Security Death Index (SSI) database inquiry was performed to determine all-cause mortality in patients who were lost to follow-up. Patients were further sub-grouped on the basis of 3D RVEF ≥ 20%. Separately, patients were sub-grouped by LVEF ≥ 20% in both ICMX and NICMX cases. A cut-off of ≥20% was chosen for the RVEF based on the results of prior studies showing significance based on Kaplan–Meier (KM) survival curves. Cumulative event rates were estimated for each subgroup using the KM analysis and were compared using the log-rank test. The 3D RV/LVEFs were compared to all-cause mortality and MACE. ICMX patients were defined using the World Health Organization (WHO) criteria. Results: From a 7000-patient CMR database, 753 heart failure patients were selected. Eighty-seven patients met WHO definition of ICMX and NICMX (43 ICMX and 44 NICMX). The study patients were followed for a median of 3 years (Interquartile range or IQR 1.5–6.5 years). The mean age of patients was 58 ± 13 years; 79% were male. In ICMX, mean 3D LVEF was 21% ± 6% and mean 3D RVEF was 38% ± 14%, while for NICMX, mean 3D LVEF was 16% ± 6% and mean 3D RVEF was 30% ± 14% (*p* < 0.005 for intra- and inter-group comparison). It should be noted that LVEF < RVEF in both groups and the ejection fraction (EF) in NICMX was less than the corresponding EF in ICMX. Overall mortality was higher in ICMX than NICMX (12/40, 30% vs. 7/43, 16%; *p* < 0.05). Patients were stratified based on both RVEF and LVEF with a threshold of EF ≥ 20% separately. RVEF but not LVEF was a significant predictor of death for NICMX (χ^2^ = 8; *p* < 0.005), while LVEF did not predict death in ICMX (χ^2^ = 2, *p* = not significant). Similarly, time to MACE was predicted by RVEF for NICMX (χ^2^ = 9; *p* < 0.005) but not by LVEF in ICMX (χ^2^ = 1; *p* = NS). Importantly, RVEF, while predictive of NICMX MACE, did not emerge as a predictor of survival or MACE in ICMX. Conclusions: Via 3D CMR in non-ischemic CMX patients, RVEF has important value in predicting death and time to first MACE while 3D LVEF is far less predictive.

## 1. Introduction

The need for improved prognostication in heart failure patients has led to a recent interest in investigating the use of right ventricular systolic function as a prognostic marker. Variables depicting right ventricle (RV) function, such as right ventricle ejection fraction (RVEF) [1,2,3,4], RV size [5,6,7] and the presence of tricuspid valve insufficiency [8], have been studied for prognostic value in heart failure. Diagnostic modalities used in these studies to measure RV function were radionuclide ventriculography, echocardiographic measurements and cardiac magnetic resonance imaging (CMR). A recent meta-analysis found that most of these studies suffer from significant between-study heterogeneity and publication bias [9]. Publication bias is a phenomenon whereby studies with certain characteristics are more likely to be accepted for publication out of all the studies conducted and submitted for publication. Namely, studies with positive results are more likely to be accepted for publication or presentation, although studies with negative results may be equally or more important with regard to the particular topic of interest. The assessment of accurate RV volumetrics is critical in establishing the predictive value of RV parameters. Assessment is difficult by virtue of the complex geometry and location of the RV. CMR is considered to be the “gold standard” for measuring RVEF. It is a non-invasive and convenient [5,10,11,12] method of assessing the RV anatomy and fibrosis pattern through late gadolinium enhancement (LGE). A PubMed search revealed a total of 20 studies that investigated the value of right-sided systolic function in predicting outcomes in heart failure patients [1,2,3,4,7,8,13,14,15,16,17,18,19,20,21,22,23,24,25,26,27]. Of note, no individual study looked into the significance of CMR-measured right ventricular variables in predicting outcomes in the subcategories (ischemic cardiomyopathy (ICMX) and non-ischemic cardiomyopathy (NICMX)) of heart failure patients. Herein, we explore the value of RVEF as an independent prognostic indicator in these subcategories with respect to survival and time to major adverse cardiovascular events (MACE). We also sought to compare the relative strength of RVEF versus left ventricle ejection fraction (LVEF) as a predictor of survival and time to MACE in these subcategories.

## 2. Materials and Methods 

We designed a retrospective cohort study using the CMR database at a single tertiary care teaching hospital containing 7000 patients enrolled from 1 January 2005 to 31 October 2012. The study was reviewed and approved by the Allegheny General Hospital Institutional Review Board (IRB No. 5631, 31 October 2012). Out of these, 753 patients had three-dimensional (3D) LVEF ≤ 35%. We excluded patients with combined nonischemic and ischemic cardiomyopathy, ordinal reporting of RVEF on the CMR report and those patients who lacked follow-up (available in the Appendix A). Forty-three patients were included in the ICMX group. World Health Organization (WHO) criteria were used to define ICMX [13]. Likewise, 44 patients were included in the NICMX group that did not have significant coronary artery disease but had 3D LVEF ≤ 35%. Significant coronary artery disease was defined as a history of prior myocardial revascularization or myocardial infarction, more than 75% stenosis in the left main stem or proximal left anterior descending artery, or two vessels or more with greater than 75% stenosis. Most patients in the NICMX group had viral, idiopathic dilated or drug-induced cardiomyopathy. Figure 1 and Figure 2 depict representative CMR images of patients with NICMX and ICMX. The subendocardial localization of a myocardial scar within a coronary territory suggests an ischemic cause, whereas other diseases have the characteristic patterns of a myocardial scar helping in the differential diagnoses of NICMX.

Patients selected for the study were tracked forward in time after their scan date by following their electronic medical record. The occurrence of either MACE or death from any cause was ascertained. MACE was defined as death, nonfatal acute coronary syndrome, nonfatal stroke or readmission for heart failure or defibrillator implantation. Basic demographic characteristics including age, gender, ejection fraction (EF), type of cardiomyopathy and duration of follow-up were determined. Additionally, vital status and MACE were recorded. Further, the time interval between the CMR scan date and death/first MACE was determined. A two-tailed non-paired Student t-test was used to compare the RVEF to LVEF within ICMX and NICMX groups. RVEF_ICMX_ vs. RVEF_NICMX_ and LVEF_ICMX_ vs. LVEF_NICMX_ analyses were run as well. Median and interquartile range (IQR) were determined for the time to MACE and overall survival. For both ICMX and NICMX groups, RVEF and LVEF were separated with the threshold of EF ≥ 20%. A cut-off of ≥20% was chosen for the RVEF based on the results gained from prior studies showing significance based on Kaplan–Meier (KM) survival curves [28]. 

Cumulative event rates were estimated using the Kaplan–Meier (KM) method and were compared using the log-rank test. Statistical analysis was performed using PASW Statistics version 18.0 (IBM SPSS, Chicago, IL, USA). Significance for the *p*-value was set at <0.05. A biomedical statistician performed the statistical review of the study. RV/LVEFs were compared linearly to MACE and time to first MACE. KM curves were used for each of the groups and EF was employed to determine if there was a significant difference in the time to MACE and time to death. Survival data were available for 95% of patients at the time of analysis. Patients in whom the alive/dead status could not be determined (5%) from the Social Security Death Index (SSI) inquiry were excluded from the time to death KM analysis. Follow-up charts for MACE were available for 78%, with the other 22% were similarly excluded from the KM analysis.

## 3. Results

Table 1 shows the demographic characteristics of the study population in both groups. The percentage of males was 90% in ICMX and 68% in NICMX. Both the 3D RVEF and the 3D LVEF were significantly lower in the NICMX group compared to the ICMX group (Figure 3). Within both the groups, the mean RVEF was significantly higher than the mean LVEF. Over a median follow-up of 3 years (IQR 1.5–6.5 years), 7/43 (16%) patients in the NICMX group died compared to 12/40 (30%) in the ICMX group (Figure 4). In the NICMX and ICMX groups, 12/32 (37%) and 18/36 (50%) experienced MACE over a median follow-up of 13 months (Table 2). At 13 months, patients with NICMX and an RVEF ≥ 20% (*n* = 34) had 19 events (i.e., MACE without death); patients with an RVEF < 20% (*n* = 10) had nine events (56% vs. 90%). The log-rank test shows a statistically significant difference between groups (χ^2^ = 7.3, df = 1, *p* = 0.007). Patients with NICMX and an LVEF ≥ 20% (*n* = 13) had nine events (i.e., MACE without death); patients with an LVEF < 20% (*n* = 31) had 19 events (69.2% vs. 61.3%). The log-rank test shows no statistically significant difference between groups (χ^2^ = 0.09, df = 1, *p* = 0.76). Patients with NICMX and an RVEF ≥ 20% (*n* = 33) had two events (i.e., MACE without death); patients with an RVEF < 20% (*n* = 8) had three events (6.1% vs. 37.5%). The log-rank test shows no statistically significant difference between groups (χ^2^ = 10.5, df = 1, *p* = 0.001).

Importantly, freedom from MACE and death at 13 months in NICMX patients was 90% in the RVEF ≥ 20% patients, compared with 60% in RVEF < 20% patients (χ^2^ = 7.3, df = 1, *p* ≤ 0.01). Among NICMX patients with LVEF ≥ 20%, the MACE rate was 61% compared with 69% in patients with LVEF < 20% (χ^2^ = 0.093, df = 1, *p* = 0.76). Freedom from death at 13 months in NICMX patients was 37.5% in the RVEF ≥ 20% patients, compared with 6.1% in RVEF < 20% patients (χ^2^ = 10.5, df = 1, *p* = 0.001).

## 4. Discussion

Heart failure (HF) is associated with a 1-year mortality of 24–28% [29,30,31]. Mortality at 5 years is 45–60%. The role of RV in helping predict short-term [12] as well as long-term [7,13,32] outcomes in heart failure has been well studied. Many studies have documented the significance of RVEF in predicting survival, hospitalization for worsening HF and non-fatal myocardial infarction (MI) in patients with ischemic heart disease [1,16,17,22,27,28,33]. Of note, most of the other studies investigating the prognostic capability of EF have not used CMR derived-RVEF for measuring outcomes in NICMX as a separate subgroup of the overall class of heart failure patients. Our study specifically sought to investigate the prognostic value of 3D RVEF in NICMX and ICMX.

While RVEF can be quantitated by CMR as well as other methods including echocardiography, radionuclide imaging and computed tomography (CT) angiography, developments in CMR image acquisition and processing have improved the quality of images and hence the accuracy of estimating true 3D RV dimensions via CMR. CMR is recognized as the reference standard for RVEF. The intra-observer and inter-observer reproducibility of RV has demonstrated the superiority of CMR over other methods in measuring RV parameters [10,11]. Therefore, CMR is considered to be the preferred modality for measuring RV function in HF.

There are many predictive models and risk scores that combine various predictors of survival in heart failure. Surprisingly, none of these include right ventricular parameters [32]. Since there is a potential role for accurate 3D RVEF in predicting MACE in non-ischemic cardiomyopathy patients, there is an obvious opportunity to include RVEF as measured by CMR in an integrated risk score model that would add value above the traditional factors such as the New York Heart Association (NYHA) class, peak O_2_ consumption, LVEF and the presence of other co-existing morbidities including chronic kidney disease and chronic obstructive pulmonary disease. This is the premise for our study and, likewise, our chief finding that 3D RVEF is a strong predictor of death and other MACE in NICMX. Surprisingly, this predictor is markedly superior compared to LVEF.

To our knowledge, we have shown for the first time that RVEF is superior to LVEF in predicting prognosis in patients with NICMX. LVEF has previously been shown to have significant limitations when used as a univariate predictor in patients with heart failure. Interestingly, in our study population, LVEF was not predictive of MACE in patients with ICMX. Inherent difference in the physiology of patients with NICMX as compared to ICMX might explain our observation of the greater predictive value of RVEF.

### Limitations

There are several limitations in our study. The occurrence of MACE was predominantly studied within the same hospital system. This could lead to selection bias. Overall, about 22% patients could not be included in the KM analysis for the time to first non-death MACE. However, vital status was determined with SSI, giving a more complete follow-up. Possible confounding factors such as age, gender and LVEF were not controlled while stratifying patients on the basis of EF, given the small sample size.

## 5. Conclusions

CMR indicating RVEF ≥ 20% has an independent predictive value for overall, as well as event-free, survival in patients with non-ischemic cardiomyopathy. However, LVEF failed to reach statistical significance as an independent predictor of either survival or time to MACE in either ICMX or NICMX groups. This finding should shed new insight and might attest to the accuracy of cardiac MRI to define RV metrics that were historically poorly performed by low-resolution imaging such as echocardiography, MUGA (multi-gated acquisition scan) and/or cardiac catheterization.

Our study also highlights the limitations of using EF alone for prognostication. The NICMX group had a statistically significantly lower EF for both ventricles compared to the ICMX group. However, the number of patients who died and/or experienced MACE in the NICMX group was numerically lower than those in the ICMX group. This further underscores the need to develop robust risk scores that incorporate a variety of factors that influence outcomes in heart failure patients. In the meantime, CMR-derived 3D RVEF is a worthy metric to further prognosticate patients presenting with HF.

## Figures and Tables

**Figure 1 diagnostics-09-00016-f001:**
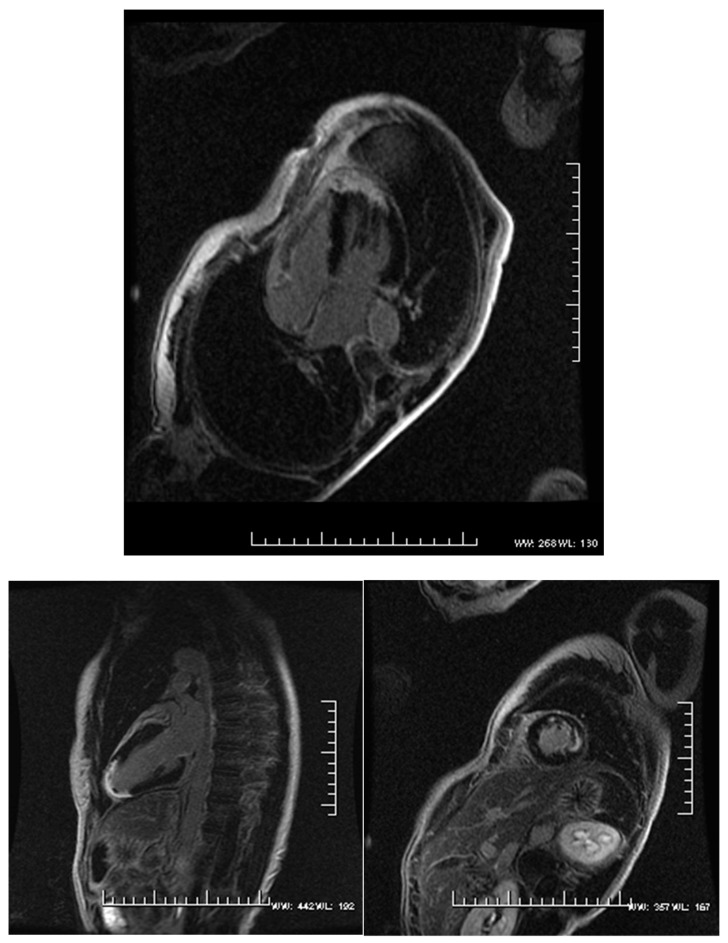
Cardiac MRI (CMR) late gadolinium enhancement (LGE) images (top, 4-chamber; bottom left, 2-chamber; and bottom right, short axis) depicting ischemic cardiomyopathy and late gadolinium enhancement (LGE) evidence of infarct pathology over the left ventricular apex in a patient with ischemic cardiomyopathy (ICMX). The depth, location and global conformity to known coronary artery territories help with the diagnosis.

**Figure 2 diagnostics-09-00016-f002:**
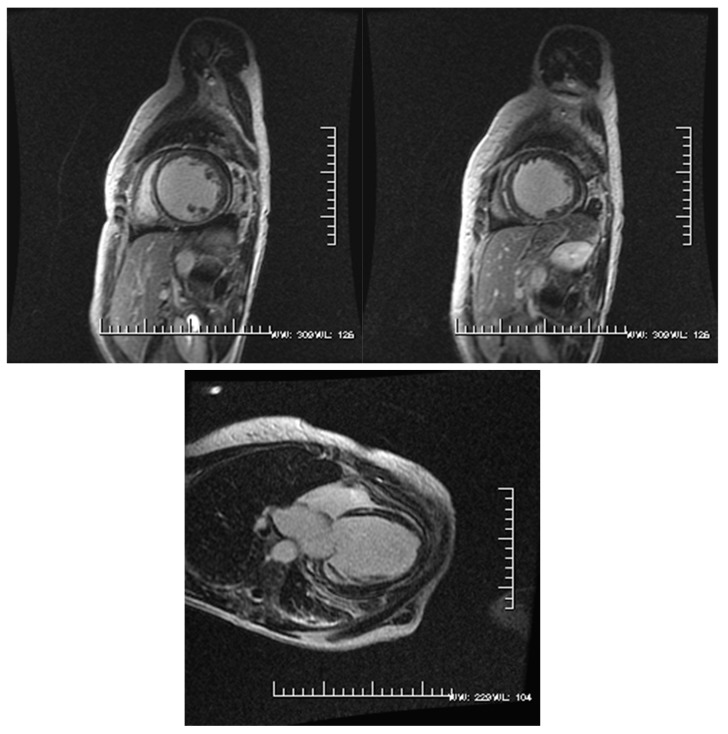
Cardiac magnetic resonance (CMR) late gadolinium enhancement (LGE) images (top row, short axis and bottom, 4-chamber) depicting the myocardial stripe (bright, gray area in the myocardium) and a dilated left ventricle (LV) in a patient with nonischemic cardiomyopathy (NICMX).

**Figure 3 diagnostics-09-00016-f003:**
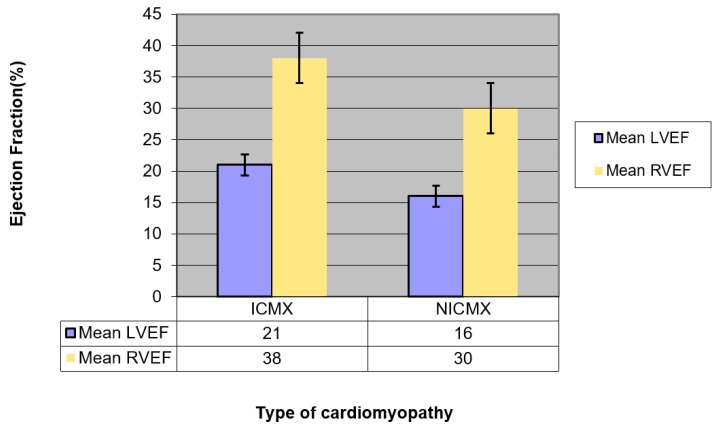
Graphical representation of the mean EF and SEM for the right ventricle (RV) and left ventricle (LV) in ICMX and NICMX groups. Ejection fractions were significantly greater in ICMX—*p* = 0.001 (LVEF _ICMX_ >> LVEF _NICMX_), *p* = 0.007 (RVEF _ICMX_ >> RVEF _NICMX_)—although they had a poorer survival. Also, RVEF > LVEF (*p* < 0.05) in both ICMX and NICMX subgroups.

**Figure 4 diagnostics-09-00016-f004:**
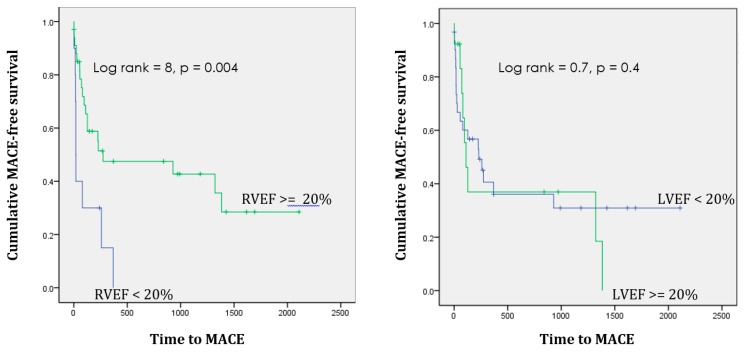
Kaplan–Meier (KM) analysis for event-free survival in patients with NICMX separated by an ejection fraction ≥20%. The *X*-axis is the time to a major adverse cardiovascular event (MACE), shown in days. The *Y*-axis is cumulative MACE-free survival. It is shown from 1.0 (depicting 100% survival) to 0.0 (depicting 0% survival). The left panel shows the stratified right ventricle ejection fraction (RVEF), and the right panel shows the left ventricle ejection fraction (LVEF) stratified by ≥20%. The green lines represent ejection fractions (EFs) ≥ 20% and the blue lines represent EFs < 20%. Log rank statistics comparing the high EF group vs. the low EF group in terms of MACE-free survival are shown along with respective *p*-values.

**Table 1 diagnostics-09-00016-t001:** Demographics for our study population.

Parameter	Ischemic CMX	Non-Ischemic CMX	Significance
**Number of patients**	43	44	
**Mean age ± SD (years)**	65 ± 11	51 ± 12	*p* < 0.05
**Percentage of males**	90%	68%	
**Mean LVEF (95% confidence interval (CI))**	21 (19.2–22.8)	13 (11.2–14.8)	*p* = 0.001 (LVEF _ICMX_ >> LVEF _NICMX_)
**Mean RVEF (95% CI)**	38 (33.8–42.2)	30 (25.8–34.1)	*p* = 0.007 (RVEF _ICMX_ >> RVEF _NICMX_)

**Table 2 diagnostics-09-00016-t002:** Comparing various parameters amongst the two groups.

Parameter	Ischemic CMX	Non-ischemic CMX	Significance
**Patients lost from MACE analysis**	10 (24%)	4 (10%)	*p* = ns
**Median, range of follow-up for MACE (days)**	440, 2–2615	574, 3–2139	*p* = ns
**Median, interquartile range (IQR) between CMR scan date and first MACE (days)**	41, 2–91	64, 19–127	*p* = ns
**Number of patients who had MACE**	18 (42%)	12 (27%)	*p* = ns
**Pts lost from death analysis (% of total)**	3 (7%)	1 (2%)	*p* = ns
**Median, range of death follow-up (days)**	1235, 39–2612	1098, 58–2238	*p* = ns
**Number of patients who died**	12/40 (30%)	7/43 (16%)	

ns: not significant.

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
