# Peer review of "Can 3D RVEF be Prognostic for the Non-Ischemic Cardiomyopathy Patient but Not the Ischemic Cardiomyopathy Patient? A Cardiovascular MRI Study"

_diagnostics, 2019, doi:10.3390/diagnostics9010016_

Reviewer 1 Report

In this study, the authors evaluate the prognostic value of RVEF measured at the MRI,  in a population of patients with heart failure. Although the study is of some scientific interest, some points need to be clarified:
1) The authors make a distinction between ischemic and non-ischemic dilated cardiomyopathy based on the old WHO classification (which do not provide reference) and not on the more recent one proposed by Felker ( Felker GM, Shaw LK, O'Connor CM.J Am Coll Cardiol 2002 Jan 16; 39 (2): 210-8. A standardized definition of ischemic cardiomyopathy for use in clinical research). Why this choice?
2) The authors state that the study population was selected from a population of 7,000 patients undergoing cardiac MRI. It is to be deduced that at their center they made about 6200 cardiac MRIs i patients with HFpEF / HFmrEF (i.e. cardiomyopathies)? I find this data strange.
3) The table with population data is not reported in the manuscript.
4) The choice of RVEF cutt off (ie less than 20%) seems unusual because in most of the previous studies a cut-off <45% is used.

5)The authors state that the EF of the left ventricle has no prognostic value in patients with ischemic dilated cardiomyopathy. As this data is in conflict with the current international guidelines, please clarify this statement.

6) Please renumber the figures.

7)Please correct supplementary data, authors contributiona and founding section.

Author Response

Response to Reviewer 1 Comments 

Point 1: The authors make a distinction between ischemic and non-ischemic dilated cardiomyopathy based on the old WHO classification (which do not provide reference) and not on the more recent one proposed by Felker ( Felker GM, Shaw LK, O'Connor CM.J Am Coll Cardiol 2002 Jan 16; 39 (2): 210-8. A standardized definition of ischemic cardiomyopathy for use in clinical research). Why this choice? 

Response 1: The Felker paper investigates survival using various methods of defining ischemic cardiomyopathy. It specifically recommended to exclude patients with single vessel obstructive coronary artery disease and no history of myocardial infarction or revascularization from ischemic category. The definition used in our paper for ischemic cardiomyopathy is consistent with the Felker paper. The particular reference for that is number 13 but was marked as number 28 by mistake on the original draft submission. This has been corrected.  

Point 2: The authors state that the study population was selected from a population of 7,000 patients undergoing cardiac MRI. It is to be deduced that at their center they made about 6200 cardiac MRIs i patients with HFpEF / HFmrEF (i.e. cardiomyopathies)? I find this data strange. 

Response 2: Cardiac MRI database at our institution included several patients whose MRI LVEF fell above 35%. This LVEF may differ from the echocardiographic LVEF. Secondly, a large proportion of patients were excluded as their RVEFs were not numerically calculated during the original image acquisition.  

Point 3: The table with population data is not reported in the manuscript. 

Response 3: Table with population data is now added.  

Point 4: The choice of RVEF cutt off (ie less than 20%) seems unusual because in most of the previous studies a cut-off  << span="">45% is used.  

Response 4: The RVEF cutoff was based on previous paper by Meyer (Meyer P, Filippatos GS, Ahmed MI, Iskandrian AE, Bittner V, Perry GJ, White M, Aban IB, Mujib M, Dell'Italia LJ, Ahmed A. Effects of right ventricular ejection  fraction on outcomes in chronic systolic heart failure. Circulation. 2010 Jan 19;121(2):252-8). This study showed significant prognostic implication of RVEF using a cutoff of less than 20%. 

Point 5: The authors state that the EF of the left ventricle has no prognostic value in patients with ischemic dilated cardiomyopathy. As this data is in conflict with the current international guidelines, please clarify this statement. 

Response 5: This finding from our paper is based on the most accurate estimation of LVEF and RVEF in our study population. Prior reports have used left ventricle ejection fraction estimated by other modalities as the sole prognostic marker with marked limitations noted in that strategy. Our paper highlights the importance of RVEF as a large contributor to prognosis in patients with heart failure with reduced ejection fraction secondary to non-ischemic cardiomyopathy. 

Point 6: Please renumber the figures.  

Response 6: Figures renumbered, thanks 

Point 7: Please correct supplementary data, authors contributiona and founding section.  

Response 7: Supplementary data, authors contribution and funding section is now updated.  

Reviewer 2 Report

The authors presented a study to evaluate CMR-derived 3D RVEF as prognostic for the non-ischemic Cardiomyopathy patient. The article is well-written and easy to read. The introduction is informative. Study is designed properly and systematically. Test results supported the conclusion of the study. The findings might be useful to prognostication in heart failure patients and inspire other researchers. Some suggestions which may further improve the article:

1. It is better to use [] instead of () for reference. () is mostly for equation.

2. The authors can consider adding more physiological explanations based on their findings. For example, why RVEF is better than LVEF in predicting prognosis in patients.

Author Response

Response to Reviewer 2 Comments 

Point 1: It is better to use [] instead of () for reference. () is mostly for equation.

Response 1: An open source citation software (Zotero) was used in preparation of this manuscript. Given the settings in Zotero, it would not allow for changes to be made in the type of parenthesis used in the text. 

Point 2: The authors can consider adding more physiological explanations based on their findings. For example, why RVEF is better than LVEF in predicting prognosis in patients.

Response 2: Although the exact explanation of these findings is unknown, we have mentioned structural differences (geometry) and more accurate calculation of function for the first time by cardiac MRI among the possible reasons for why we are now starting to see this superior prognostic value of RVEF over LVEF. Additionally, other textbooks have postulated that relative differences in the wall thickness and diffusion of nutrients from chamber cavity as a contributor to resilience to ischemic insults in the right ventricle making it more susceptible to non-ischemic insults as well.

We are thankful to the reviewer for their kind suggestions and comments. 

Round  2

Reviewer 1 Report

The reviewer thank the authors for the reply and has no further questions.